# Fractal dimension analysis of different mandibular regions in familial Mediterranean fever patients: A cross-sectional retrospective study

**Nilüfer Ersan**[ID][1]*, **Beliz Özel**[ID][2¤]

1 Yeditepe University Faculty of Dentistry, Department of Dentomaxillofacial Radiology, Istanbul, Turkiye,
2 Yeditepe University Faculty of Dentistry, Department of Endodontics, Istanbul, Turkiye

¤ Current address: Department of Endodontics, Academic Centre for Dentistry Amsterdam (ACTA), Vrije Universiteit Amsterdam, Amsterdam, Netherlands
* yasenil@yahoo.com

## Abstract

Familial Mediterranean fever (FMF) is a genetic condition that may cause loss of bone mineral density (BMD) due to chronic inflammation. Previously, fractal dimension (FD) analysis values of mandibular cortical bone were shown to be lower in osteoporosis. Therefore, FD might be considered as an auxiliary tool to refer patients for dual-energy x-ray absorptiometry (DXA), which is the gold standard for BMD measurement. The purpose of this cross-sectional retrospective study was to evaluate trabecular and cortical microarchitecture of the mandible with FD analysis on panoramic radiographs in a subpopulation of FMF. Also, the effect of colchicine use was investigated. Forty-three FMF patients, aged between 10.8 and 71.2 years, and age- and gender-matched control group consisting of patients, who had no systemic diseases, were included. Demographic information such as age and gender, and colchicine use were recorded. In terms of age, the patients were classified as <30 and 30< years. On each panoramic radiographs five regions of interest were selected on the mandible as: 1- premolar, 2- molar, 3- angular, 4- condylar, and 5- basal cortical bone regions on right (R) and left (L) sides. Statistical significance was accepted at p<0.05 level. Intra- and inter-observer agreements demonstrated good to excellent consistency. In FMF patients, L3 and L4 values were higher, whereas L5 values were lower (p<0.05) than the control group. In terms of age, the difference between groups was insignificant in FMF patients (p>0.05), whereas in control group R3 and L4 values were higher in the 30< age group (p<0.05). Regarding gender and colchicine use, the difference between groups was insignificant (p>0.05). FMF disease might be a candidate for referral to DXA examination based on decreased bone density in the mandibular cortex detected by FD measurements on routine panoramic radiographs. Further studies are warranted to ascertain this relationship.

**Data Availability Statement:** All relevant data are within the paper and its Supporting Information files. Additional data is available from the figshare repository (http://doi.org/10.6084/m9.figshare.23259770).

**Funding:** The authors received no specific funding for this work.

**Competing interests:** The authors have declared that no competing interests exist.

## Introduction

Familial Mediterranean fever (FMF), which was described by Heller in 1955, is among the most common monogenic autoinflammatory diseases [1]. The disease most commonly affects Jewish, Armenian, Turkish, and Arab populations [2]. Mutation of the *MEFV* gene was shown to be responsible for the development of FMF. The clinical hallmark of FMF is recurring fever attacks in relation with serositis with a frequency varying between weekly to once in several months that mostly takes 1–3 days that is followed by a spontaneous resolving [3]. The patient is mostly asymptomatic between febrile attacks, even though it was reported that subclinical inflammation continues between attacks [4,5].

Colchicine, which is the main drug used in the treatment of FMF, was approved in the United States by the Food and Drug Administration in 2009. Colchicine is mainly used to prevent FMF acute attacks caused by its several anti-inflammatory effects. It may as well prevent amyloidosis, which arises secondary to the chronic inflammatory conditions, and could also be seen in bones [4,6]. Although colchicine is considered to be a lifelong treatment for FMF, a recent study demonstrated that some physicians decide to discontinue the drug in patients, who were carriers of a pathogenic variant or variants of unknown significance and did not experience any attacks at least for six months [7,8]. As well, some of the patients needed to discontinue the drug due to colchicine resistance or adverse effects of the drug.

Osteoporosis is characterized by decreased bone mass and microarchitectural deterioration of bony tissue that may subsequently lead to increased fragility and bone fracture [9]. Chronic subclinical inflammation in FMF patients might also give rise to decreased bone mineral density (BMD) and osteoporosis by affecting bone turnover and metabolism [10,11]. Many authors showed low levels of BMD and bone formation markers in individuals with FMF [11–13]. Colchicine decreases osteoclast numbers and inhibits resorption in bones [14,15], and therefore improve bone density and prevent osteoporosis [4,11,13].

Diagnosis of osteoporosis at an early stage is an important healthcare issue. The current principal method for diagnosing osteoporosis is measurement of BMD by dual energy X-ray absorptiometry (DXA) [16]. DXA is the gold standard for BMD measurements and is used for the diagnosis of osteopenia and osteoporosis, as well as prediction the fracture risk. The distribution of the equipment and cost of advanced imaging techniques, such as DXA, limit their access for screening of larger populations [17].

Another method to determine the changes related to bone mineral loss is fractal dimension (FD) analysis on panoramic radiographs. Panoramic radiography is one of the most used imaging modalities in dentistry and is an important part of routine dental care. It provides a valuable screening opportunity due to its lower cost and radiation dose, as well as a high access. Fractal analysis, which is a mathematical image analysis method, is used in the analysis of complex shapes and structural formations. The interpretation of the mandible on a panoramic image via computational analysis methods gives objective numerical results defined as the FD, and thus rules out the subjective judgment of the observer [18]. FD can assist in the quantification of complex structures and description of bone microarchitecture and demonstrate bone mineral loss on panoramic images [19]. And thus, FD is considered to be a distinctive parameter for the determination of the density of the osteoporotic and normal bone tissue [20,21]. The increased FD values have been linked to the increased complexity of the structure. A recent systematic review concluded that FD analysis on dental images is a reliable diagnostic tool for osteoporosis screening and could be a reference BMD test [22]. Even though both trabecular and cortical osseous tissue have a fractal structure, this structure is not visible to the naked eye for cortical bone [23]. In previous studies mostly trabecular bone regions were selected for the FD evaluation, while there are also studies including the cortical bone areas, as

well [20,24–32]. Sindeaux et al. [30] demonstrated that FD values of the cortical bone were more accurate than those of the trabecular bone. They also reported that the patients with osteoporosis have higher probability of having lower mean FD values on the cortical bone compared to healthy counterparts and that cortical bone FD measurements might be considered as auxiliary tools to refer patients for DXA exam [30].

To the best of our knowledge, there is only one study that evaluates the bone microarchitecture of FMF patients using FD analysis that was performed on the trabecular structure of the mandible via panoramic radiographs of solely pediatric patients [33]. To date, no study exists, in which FD analysis was applied on panoramic radiographs of adult FMF patients, taking the colchicine use and mandibular cortical structure into consideration, as well. The hypothesis was that there were no differences in the microarchitecture of the mandible between FMF and healthy populations. Therefore, the aim of this cross-sectional retrospective study was to assess trabecular and cortical bone structures on panoramic radiographs in a subpopulation of FMF in order to find out whether significant differences exist in FMF patients and healthy individuals. Also, the effect of colchicine use on mandibular cortical and trabecular bone microarchitecture was investigated in FMF patients.

## Materials and methods

All the procedures followed were in accordance with the ethical standards of the Helsinki Declaration and was approved by the Yeditepe University Non-interventional Research Ethics Committee (approval number: 202208Y0279). Written informed consent was obtained from the patients and/or their legal guardian at the time of panoramic imaging for the exposure and also for the possible use of the images for scientific purposes. However, for the study the requirement for patients' informed consent was waived because of the retrospective nature of the study. All panoramic images were fully anonymized before the data collection and used in accordance with the relevant guidelines and ethical regulations. The individual in this manuscript has given written informed consent to publish these case details. Sample size calculation performed by G* Power 3.1.9.2 (Kiel, Germany) revealed that a total of minimum 50 panoramic radiographs from FMF and control patients (n = 25 in each group) was necessary (confidence interval: 95%, significance level: 0.05, effect size (d): 0.47).

The database of the Department of Dentomaxillofacial Radiology of the Yeditepe University Faculty of Dentistry was retrospectively reviewed in August-September, 2022. Overall 75 self-reported FMF patients, who underwent panoramic imaging at Yeditepe University Faculty of Dentistry between January 2014 and November 2019 and have a panoramic radiograph with a good image quality, were identified. Among these patients, 32 of them, who had systemic diseases that affect bone metabolism (15 patients), and were using medications affecting the bone metabolism (3 patients), as well as the patients with a mixed dentition (13 patients) and severely atrophic alveolar crest (1 patient) that complicated the measurements on panoramic radiographs, were excluded. As a result, 43 FMF patients aged between 10–71 years and a control group consisting of the same number of randomly selected age-gender-matched patients, who had no systemic diseases, were included in the study. Control group, which was also subjected to the exclusion criteria, was selected randomly from the patients in the same database, depending on the matching characteristics of the patients in the FMF group, in terms of age and gender by filtering the data on Microsoft Excel sheet (Microsoft Office Professional Plus 2010 v14.0, Microsoft Corporation). The remaining patients were assigned a number consequently and control patients corresponding to the FMF patients were randomly selected by using a random number generator (random.org). Demographic information such as age and gender, and colchicine use were recorded. The patients were further classified into two age

groups as <30 and 30< years. Besides, the patients with FMF were classified according to colchicine use. Self-reported colchicine use was recorded as 'Yes' or 'No', without taking the prior use or dose regimen into consideration. A flow chart of the study design demonstrating the eligibility and the numbers of individuals recruited at each stage of study (Fig 1).

A total of 86 panoramic radiographic images obtained with Planmeca 2002 cc Proline (Planmeca, Helsinki, Finland; 70 kVp, 10 mA, 8 s exposure time) and x1.34 magnification factor, were evaluated. Orientation of the head was arranged so that the Frankfort horizontal plane was parallel to the floor and the sagittal plane was parallel to the vertical plane. Digital images were exported in 8-bit depth grayscale high resolution '.tiff' format from the Planmeca Romexis 3.8.3 (Helsinki, Finland).

The FD measurements were performed on the ImageJ software (ImageJ 1.38; US National Institutes of Health, Bethesda, MD, USA) that was downloaded from https://imagej.nih.gov/ij/download.html. All sets of the fully anonymized panoramic images were imported into ImageJ software. Four different regions of interest (ROI) in 30x30 pixel size were selected from designated spots (rectangle tool) on right (R) and left (L) side of the mandible as follows (Fig 2); ROI1: Distal region of the premolar, next to the mental foramen, ROI2: mesial region of the apical part of second molar, ROI3: angular region of the mandible, and ROI4: mandibular condyle area. An additional fifth ROI was selected in differing pixel size depending on the ROI (polygon tool) on each side of the mandible as; ROI5: basal cortical bone of the mandible extending from distal to the mental foramen to the distal root of the first molar (Fig 2). The individual in this manuscript has given written informed consent to publish these case details. Each ROI was selected and the FD analysis was conducted as follows; duplication of the selected ROI, application of the Gaussian filter [34] to remove brightness alterations due to overlying soft and hard tissue, subtraction of the filtered image from the original cropped image, addition of a gray value of 128 to differentiate bone marrow spaces and trabeculae, binarization of the resulting image, steps of erosion, dilatation, inversion, and skeletonization. Lastly, the FD was calculated according to the fractal box counting method described by White & Rudolph [34]. The FD analysis was carried out by two independent observers (a dentomaxillofacial radiology specialist with 12-year of experience and an endodontist with an 8-year of experience). Prior to the FD analysis, the observers were calibrated by evaluating 15 panoramic radiographs, which were not included in the study, together. After the first independent readings 25% of the measurements were repeated after a two-week interval for intra- and interobserver repeatability. Data collection was completed in November, 2022. FD analysis on panoramic radiographs was comparable for both FMF and control groups that all the fully anonymized panoramic images were evaluated in the same manner by two observers. The observers had access to patient information that could identify individual participants after data collection.

The fact that the information regarding the diagnosis of FMF and colchicine use, as well as other possible diseases and drug use questioned during taking anamnesis and evaluated in the exclusion criteria was based on the self-reported information was a potential confounder and a limitation of the study. Age was another confounder that we tried to overcome by selecting age and gender matched patients in the control group. Also, we classified the patients in two age groups as <30 and 30< years.

SPSS software version 25.0 (IBM, USA) was used for the statistical analysis. Normality distribution of all variables was analyzed with histogram graphics and Kolmogorov-Smirnov tests. Mean, standard deviation, median, and IQR values were used for descriptive analysis. Analysis of the nonparametric variables that did not display a normal distribution among two groups was performed with Mann-Whitney U test, whereas Independent t-test was performed to analyze parametric variables that were normally distributed. Chi-square test was used to

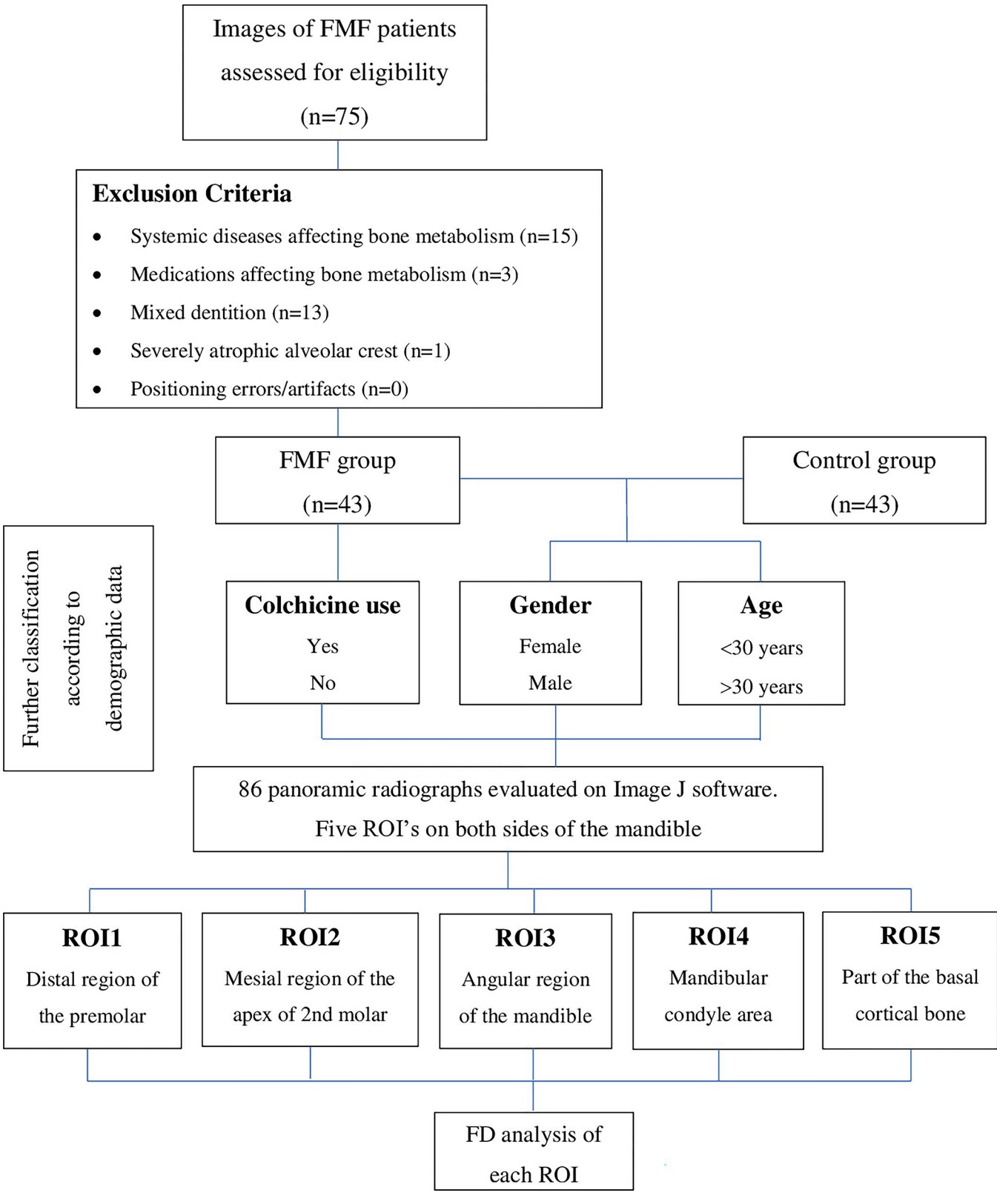

**Fig 1. Flow chart of the study design.**

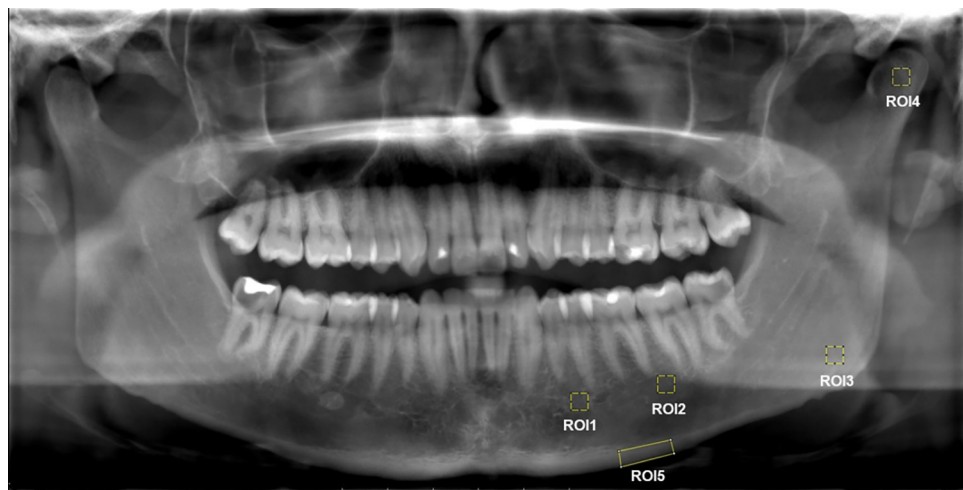

**Fig 2. Designated region of interests (ROI) indicated on the panoramic radiograph.**

assess the distribution of categorical data between FMF and control groups. Intra- and inter-observer agreement was assessed with Intraclass Correlation Coefficient (ICC) [35]. Statistical significance was accepted at p<0.05 level.

## Results

Demographic distribution of the patients is demonstrated on Table 1.

Regarding all the ROI measurements, intra-observer agreement of the first and second observer, and inter-observer agreement that were determined using ICC test demonstrated a good to excellent agreement that ranged between 0.752–0.944, 0.802–0.958, and 0.765–0.962, respectively (p<0.05, Table 2).

Comparison of the FMF and control groups regarding right and left ROI measurements revealed that in FMF patients mean L3 and L4 values were found to be higher, whereas mean L5 values were less than the control group (p<0.05, Table 3). In terms of age groups, the difference between ROI measurements on both sides in FMF group was not significant (p>0.05), whereas in control group R3 and L4 values were found to be higher in 30< age group comparing to <30 age group (p<0.05, Table 4). In terms of gender in both groups and colchicine use

**Table 1. Demographic distribution of the FMF and control patients.**

|  |  | FMF | Control | Total | P- value |
|---|---|---|---|---|---|
|  |  | **Mean±SD**<br>**Median (min-max)** | **Mean±SD**<br>**Median (min-max)** | **Mean±SD**<br>**Median (min-max)** |  |
| **Age** |  | 32.1±12.8<br>31.3 (10.8–71.2) | 35.0±15.1<br>30.5 (11.3–70.7) | 33.6±14.1<br>30.9 (10.8–71.2) | 0.354[1] |
|  |  | **n (%)** | **n (%)** | **n (%)** | **P- value** |
| **Gender** | **Female** | 27 (62.8) | 32 (74.4) | 59 (68.6) | 0.352[2] |
|  | **Male** | 16 (37.2) | 11 (25.6) | 27 (31.4) |  |
| **Colchicine use** | **No** | 13 (30.2) | NA | 13 (30.2) | *** |
|  | **Yes** | 30 (69.8) | NA | 30 (69.8) |  |

[1] Independent t-test

[2] Chi-square test p<0.05 NA: not available.

**Table 2. Intra- and inter-observer agreements determined using ICC test.**

|  | 1st observer 1st & 2nd reading | | 2nd observer 1st & 2nd reading | | 1st & 2nd observer 1st readings | | 1st & 2nd observer 2nd readings | |
|---|---|---|---|---|---|---|---|---|
|  | ICC | P- value | ICC | P- value | ICC | P- value | ICC | P- value |
| R1 | 0.920 | <0.001* | 0.811 | 0.010* | 0.897 | 0.001* | 0.865 | 0.003* |
| R2 | 0.915 | 0.001* | 0.802 | 0.012* | 0.831 | 0.007* | 0.881 | 0.002* |
| R3 | 0.921 | <0.001* | 0.936 | <0.001* | 0.962 | <0.001* | 0.962 | <0.001* |
| R4 | 0.902 | 0.001* | 0.879 | 0.002* | 0.912 | 0.001* | 0.807 | 0.011* |
| R5 | 0.885 | 0.002* | 0.900 | 0.001* | 0.765 | 0.021* | 0.853 | 0.004* |
| L1 | 0.867 | 0.003* | 0.805 | 0.012* | 0.868 | 0.003* | 0.877 | 0.002* |
| L2 | 0.910 | 0.001* | 0.868 | 0.003* | 0.934 | <0.001* | 0.937 | <0.001* |
| L3 | 0.752 | 0.026* | 0.958 | <0.001* | 0.951 | <0.001* | 0.917 | 0.001* |
| L4 | 0.804 | 0.012* | 0.952 | <0.001* | 0.884 | 0.002* | 0.864 | 0.003* |
| L5 | 0.944 | <0.001* | 0.907 | 0.001* | 0.869 | 0.003* | 0.841 | 0.006* |

Intraclass Correlation Coefficient *p<0.05.

in the FMF group, the difference between ROI measurements on right and left sides was insignificant (p>0.05, Tables 5 and 6).

## Discussion

Fractal dimension analysis has been performed for the determination of the architectural structure of the trabecular and cortical bone on panoramic radiographs [20,24–32]. The

**Table 3. Comparison of fractal dimension measurements obtained on right and left sides on the panoramic radiographs of FMF and control groups.**

|  | FMF (n = 43) mean±SD median (IQR) | Control (n = 43) mean±SD median (IQR) | P- value |
|---|---|---|---|
| R1 | 1.241±0.117 1.262 (1.191–1.314) | 1.260±0.089 1.255 (1.204–1.339) | 0.880[1] |
| R2 | 1.277±0.076 1.299 (1.229–1.328) | 1.243±0.111 1.255 (1.176–1.332) | 0.134[1] |
| R3 | 1.227±0.113 1.226 (1.152–1.301) | 1.246±0.112 1.255 (1.156–1.350) | 0.458[2] |
| R4 | 1.221±0.134 1.250 (1.159–1.301) | 1.201±0.104 1.218 (1.120–1.279) | 0.443[2] |
| R5 | 1.115±0.094 1.119 (1.059–1.172) | 1.130±0.096 1.133 (1.073–1.170) | 0.458[2] |
| L1 | 1.234±0.117 1.255 (1.159–1.314) | 1.182±0.163 1.195 (1.093–1.314) | 0.094[2] |
| L2 | 1.254±0.105 1.271 (1.168–1.323) | 1.208±0.121 1.217 (1.129–1.314) | 0.129[1] |
| L3 | 1.236±0.119 1.240 (1.166–1.301) | 1.144±0.171 1.147 (1.021–1.286) | 0.013[1]* |
| L4 | 1.273±0.131 1.264 (1.184–1.342) | 1.162±0.154 1.214 (1.090–1.293) | 0.002[1]* |
| L5 | 1.110±0.072 1.105 (1.052–1.165) | 1.154±0.076 1.160 (1.089–1.221) | 0.007[2]* |

[1]Mann Whitney U test

[2]Independent t-test

*p<0.05.

**Table 4. Fractal dimension measurements in the FMF and control groups according to different age groups.**

| | FMF | | P- value | Control | | P- value |
|---|---|---|---|---|---|---|
| | **Age<30 (n = 20)** | **Age>30 (n = 23)** | | **Age<30 (n = 21)** | **Age>30 (n = 22)** | |
| | **mean±SD**<br>**median (IQR)** | **mean±SD**<br>**median (IQR)** | | **mean±SD**<br>**median (IQR)** | **mean±SD**<br>**median (IQR)** | |
| **R1** | 1.240±0.126<br>1.258 (1.167–1.326) | 1.241±0.112<br>1.262 (1.191–1.312) | 0.897[1] | 1.244±0.086<br>1.232 (1.187–1.306) | 1.275±0.091<br>1.277 (1.226–1.341) | 0.194[1] |
| **R2** | 1.281±0.076<br>1.303 (1.231–1.328) | 1.274±0.078<br>1.297 (1.224–1.320) | 0.689[1] | 1.220±0.112<br>1.227 (1.149–1.308) | 1.265±0.108<br>1.261 (1.182–1.365) | 0.226[1] |
| **R3** | 1.223±0.121<br>1.214 (1.159–1.324) | 1.231±0.108<br>1.257 (1.137–1.286) | 0.826[2] | 1.210±0.107<br>1.209 (1.131–1.296) | 1.280±0.108<br>1.317 (1.192–1.364) | **0.040[2*]** |
| **R4** | 1.236±0.137<br>1.234 (1.173–1.332) | 1.209±0.133<br>1.250 (1.153–1.285) | 0.520[2] | 1.180±0.111<br>1.188 (1.099–1.272) | 1.222±0.094<br>1.237 (1.201–1.304) | 0.193[2] |
| **R5** | 1.107±0.107<br>1.111 (1.056–1.176) | 1.121±0.083<br>1.131 (1.056–1.148) | 0.627[2] | 1.141±0.048<br>1.139 (1.099–1.185) | 1.120±0.127<br>1.086 (1.050–1.161) | 0.485[2] |
| **L1** | 1.222±0.127<br>1.245 (1.174–1.294) | 1.244±0.110<br>1.267 (1.158–1.314) | 0.563[2] | 1.156±0.116<br>1.149 (1.050–1.236) | 1.207±0.197<br>1.235 (1.182–1.337) | 0.306[2] |
| **L2** | 1.276±0.109<br>1.309 (1.212–1.337) | 1.235±0.100<br>1.248 (1.146–1.319) | 0.156[1] | 1.189±0.118<br>1.217 (1.072–1.292) | 1.226±0.123<br>1.236 (1.136–1.319) | 0.230[1] |
| **L3** | 1.216±0.145<br>1.232 (1.153–1.300) | 1.253±0.092<br>1.262 (1.184–1.301) | 0.430[1] | 1.098±0.175<br>1.125 (0.990–1.276) | 1.187±0.159<br>1.213 (1.036–1.328) | 0.126[1] |
| **L4** | 1.264±0.104<br>1.261 (1.172–1.341) | 1.281±0.153<br>1.264 (1.203–1.342) | 0.787[1] | 1.116±0.163<br>1.148 (1.066–1.224) | 1.206±0.134<br>1.237 (1.137–1.314) | **0.027[1*]** |
| **L5** | 1.119±0.084<br>1.105 (1.039–1.194) | 1.101±0.061<br>1.105 (1.052–1.144) | 0.430[2] | 1.154±0.075<br>1.160 (1.081–1.219) | 1.154±0.079<br>1.158 (1.090–1.221) | 0.999[2] |

[1]Mann Whitney U test

[2]Independent t-test

*p<0.05.

inferior mandibular cortical bone is among the most commonly studied regions in osteoporosis detection on panoramic radiographs. Even though the mandibular cortical index has been shown to be a useful tool for osteoporosis screening [36], it is a subjective visual assessment and has a relatively limited reproducibility [37].

The current study aims to evaluate mandibular microarchitecture of the individuals with FMF in comparison with healthy population and determine a possible relationship between the FD values gathered on five different regions, including trabecular and cortical regions of the mandibular bone, on both sides on panoramic radiographs. Additionally, we investigated this relationship among the FMF patients regarding colchicine use. To date, no studies evaluated mandibular bone morphology on adult FMF patients using FD analysis and took colchicine use and mandibular cortical bone into consideration.

In dental practice, panoramic radiography is routinely used and has been shown among the most useful tools to detect changes in the mandibular bone density, even if it is not primarily used for this purpose [21]. Therefore, panoramic radiography may be an important imaging modality to assess bone structures in individuals with FMF, as well. Previously, the effect of various systemic conditions affecting the jawbones have been evaluated on panoramic radiographs with FD. However, regarding FMF there is only one study, which has been conducted solely in children [33]. Besides, some studies demonstrated that colchicine treatment prevents osteoporosis [4,13]. Since in FMF chronic inflammation might give rise to a decreased bone density in the course of time during the disease process, it could be speculated that in adult FMF patients, especially in case they are not under colchicine treatment, FD analysis would

**Table 5. Fractal dimension measurements in the FMF and control groups according to gender.**

| | FMF | | P- value | Control | | P- value |
|---|---|---|---|---|---|---|
| | Female (n = 27) | Male (n = 16) | | Female (n = 32) | Male (n = 11) | |
| | mean±SD median (IQR) | mean±SD median (IQR) | | mean±SD median (IQR) | mean±SD median (IQR) | |
| R1 | 1.246±0.114 1.263 (1.226–1.313) | 1.230±0.126 1.243 (1.166–1.317) | 0.704[1] | 1.266±0.092 1.257 (1.212–1.340) | 1.242±0.078 1.232 (1.170–1.303) | 0.342[1] |
| R2 | 1.272±0.079 1.293 (1.224–1.328) | 1.286±0.071 1.305 (1.243–1.324) | 0.484[1] | 1.251±0.099 1.249 (1.190–1.324) | 1.218±0.144 1.255 (1.153–1.332) | 0.810[1] |
| R3 | 1.222±0.107 1.226 (1.144–1.278) | 1.236±0.125 1.238 (1.178–1.324) | 0.703[2] | 1.249±0.112 1.249 (1.163–1.355) | 1.235±0.118 1.255 (1.118–1.305) | 0.720[2] |
| R4 | 1.195±0.130 1.250 (1.120–1.285) | 1.265±0.134 1.260 (1.175–1.373) | 0.100[2] | 1.205±0.101 1.226 (1.132–1.273) | 1.190±0.116 1.204 (1.082–1.279) | 0.677[2] |
| R5 | 1.119±0.077 1.119 (1.059–1.172) | 1.107±0.120 1.117 (1.049–1.164) | 0.684[2] | 1.133±0.108 1.136 (1.059–1.181) | 1.120±0.049 1.127 (1.080–1.163) | 0.702[2] |
| L1 | 1.242±0.139 1.267 (1.158–1.329) | 1.220±0.070 1.221 (1.174–1.276) | 0.574[2] | 1.190±0.160 1.202 (1.130–1.303) | 1.157±0.177 1.147 (1.017–1.352) | 0.565[2] |
| L2 | 1.244±0.113 1.251 (1.146–1.320) | 1.272±0.091 1.307 (1.182–1.347) | 0.313[1] | 1.225±0.122 1.258 (1.144–1.318) | 1.160±0.107 1.177 (1.048–1.217) | 0.066[1] |
| L3 | 1.239±0.100 1.242 (1.172–1.296) | 1.230±0.151 1.225 (1.164–1.308) | 0.952[1] | 1.154±0.169 1.150 (1.030–1.290) | 1.113±0.183 1.125 (0.973–1.273) | 0.435[1] |
| L4 | 1.266±0.124 1.256 (1.203–1.325) | 1.286±0.146 1.301 (1.148–1.406) | 0.653[1] | 1.163±0.170 1.205 (1.101–1.299) | 1.160±0.101 1.214 (1.090–1.225) | 0.516[1] |
| L5 | 1.111±0.076 1.110 (1.033–1.165) | 1.108±0.068 1.094 (1.057–1.160) | 0.880[2] | 1.154±0.078 1.166 (1.086–1.218) | 1.155±0.074 1.160 (1.095–1.223) | 0.980[2] |

[1]Mann Whitney U test

[2]Independent t-test p<0.05.

assist in the detection of the expected changes. Not all the patients investigated in this study were under colchicine treatment, so that we could make a comparison between patients according to colchicine use. However, a significant difference was not observed in FMF patients, in terms of age groups or colchicine use. On the other hand, in terms of age, control patients revealed a greater R3 and L4 values in the patients aged 30< years. Different findings obtained on both sides might be the result of increased activity in the preferred side for chewing, asymmetrical skeleton of the face, the posture and functional relationship of the cheek, lips and tongue, and atrophy of the masticatory muscles in elderly people [38]. However, the evaluation of such instances was not the aim of the current study.

Whilst it was previously shown that continuous inflammation decreases the rate of osteoblastic activity [39], osteoporosis could be anticipated as a result of chronic inflammation in FMF populations [40]. The fact that the clinical presentation of the FMF patient appears to be normal, unless there is an inflammatory attack, is considered to be the most important distinguishing characteristic of this chronic inflammatory disease [41]. Therefore, even though the disease is under control without exacerbations, osteoporosis might still be encountered, especially in case the patient is not under colchicine treatment. Altunok Ünlü et al. [33] reported that FD analysis results of children with FMF were close to healthy counterparts, and this could be a result of regular colchicine use. This finding was in correlation with those of Bayrak et al. [21], who also did not present significantly different FD measurements in the FMF group. Nevertheless, there is not an agreement in the literature regarding the osteoporotic effects of FMF. Only a small number of studies were performed regarding the effects on bone density in FMF populations, some of which have described a decrease [20,34,39], while others reported that there were not any effects on BMD in FMF populations [42,43].

**Table 6. Fractal dimension measurements in the FMF group according to colchicine use.**

| | Colchicine use | | P- value |
|---|---|---|---|
| | Yes (n = 30) | No (n = 13) | |
| | mean±SD<br>median (IQR) | mean±SD<br>median (IQR) | |
| R1 | 1.243±0.118<br>1.265 (1.141–1.317) | 1.235±0.120<br>1.253 (1.212–1.305) | 0.682[1] |
| R2 | 1.273±0.074<br>1.274 (1.229–1.314) | 1.288±0.082<br>1.309 (1.261–1.348) | 0.368[1] |
| R3 | 1.236±0.116<br>1.251 (1.144–1.326) | 1.207±0.107<br>1.204 (1.169–1.261) | 0.445[2] |
| R4 | 1.218±0.137<br>1.226 (1.159–1.298) | 1.229±0.131<br>1.279 (1.133–1.325) | 0.816[2] |
| R5 | 1.106±0.096<br>1.111 (1.056–1.148) | 1.135±0.089<br>1.137 (1.052–1.215) | 0.352[2] |
| L1 | 1.242±0.113<br>1.256 (1.180–1.314) | 1.215±0.130<br>1.238 (1.117–1.310) | 0.507[2] |
| L2 | 1.257±0.105<br>1.263 (1.168–1.333) | 1.248±0.109<br>1.310 (1.151–1.320) | 0.976[1] |
| L3 | 1.223±0.134<br>1.228 (1.140–1.301) | 1.266±0.072<br>1.255 (1.210–1.311) | 0.298[1] |
| L4 | 1.269±0.130<br>1.260 (1.203–1.332) | 1.284±0.140<br>1.270 (1.176–1.342) | 0.841[1] |
| L5 | 1.103±0.073<br>1.104 (1.032–1.161) | 1.125±0.071<br>1.105 (1.067–1.170) | 0.379[2] |

[1]Mann Whitney U test

[2]*Independent t-test p<0.05.*

In our study, in FMF patients similar FD values were demonstrated in terms of age, gender, and colchicine use. However, while in the left angulus and condylar area FD measurements were significantly greater, the left mandibular cortical area revealed lower FD measurements in the FMF patients. Mandibular cortical bone corresponds to a region with an increased bone density. Sindeaux et al. [30] demonstrated that patients with osteoporosis have a higher probability of having lower mean values of FD on the cortical bone. In our study, L5 value, which concerns the mandibular cortical bone, was lower in the FMF patients. This result was consistent with the study of Suyani et al. [12], who also presented lower bone density in adults with FMF in the other body parts. This difference between different regions of the mandible could be attributed to the different trabecular or cortical structure of these regions. As well, continuing inflammation may cause retardation of the development of the mandible in certain regions, subsequently affecting the craniofacial growth and the distribution of the mineralized tissue [44]. Overall, it might be speculated that the conflicting results between studies could be attributed to the bone tissue and ROI, of which the BMD was investigated, evaluation methods, different demographic features of study groups, such as age and gender, the type of mutation that resulted in FMF, and differences in the dose regimen of prior or current colchicine use.

Regarding the studies performed on the mandible, Altunok Ünlü et al. [33], who investigated the effects of FMF on the mandible in children, reported a non-significantly different FD values in relation to gender of the children with FMF, which is in accordance with the current study performed on individuals, including adult FMF patients. They also reported that the highest intra-observer agreement for the FD measurements was found in the angulus area

[33]. Besides, they did not observe significant differences between the FD values of this region in terms of laterality, age and gender. Consequently, they suggested that the mandibular angulus region could be selected for diagnosis or follow up of the disease using FD analysis. Even though, in the current study one of the highest intra- and inter-observer agreements was observed also in the angular region, FD values demonstrated significant differences between certain groups. In this study, the mean age of the FMF group was found as 32.1±12.8, which constitutes mostly an adult group. Different results obtained in these studies could be due to the difference in age range of the patient populations.

The limitation of the study was that due to the retrospective cross-sectional nature of the study the gathered data depended on self-reported medical records obtained during questioning of the systemic anamnesis at a dental hospital and detailed information regarding FMF diagnosis, laboratory or imaging results of the test that are more accurately define BMD or treatment dosage and regimen, or previous colchicine use could not be obtained and evaluated. Additionally, information related to dental anamnesis such as dental status, parafunctional habits, and chewing side preference of the patients, as well as clinical findings which would affect the morphology of the bone also were not recorded and evaluated in this study.

## Conclusion

Within the limitations of the study, FD analysis performed on the mandible demonstrated contradictory results on the trabecular and cortical regions. The basal cortical bone microarchitecture in a subpopulation of FMF patients revealed a decreased bone density. On the contrary, trabecular bone in the mandibular angle and condyle regions showed an increase in bone density. FMF disease might be a candidate for referral to DXA examination based on decreased bone density in the mandibular basal cortex detected by FD measurements on routine panoramic radiographs. According to our results, the relationship between colchicine use and the microarchitecture of mandibular bone is conjectural. Further studies performed on larger patient populations, in which the type of mutation resulted in FMF, the colchicine regimen and other factors affecting mandibular bone structure, such as parafunctional habits and chewing side preferences, were taken into consideration, are necessary to confirm these findings.

## Supporting information

**S1 Checklist. STROBE statement—Checklist of items that should be included in reports of observational studies.**
(DOCX)

**S1 Protocol. Study protocol regarding fractal dimension analysis measurements.**
(DOCX)

**S1 Dataset. Data table.**
(XLSX)

## Author Contributions

**Conceptualization:** Nilüfer Ersan, Beliz Özel.

**Data curation:** Nilüfer Ersan, Beliz Özel.

**Formal analysis:** Nilüfer Ersan.

**Investigation:** Nilüfer Ersan, Beliz Özel.

**Methodology:** Nilüfer Ersan, Beliz Özel.

**Software:** Beliz Özel.

**Writing – original draft:** Nilüfer Ersan, Beliz Özel.

**Writing – review & editing:** Nilüfer Ersan.

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
