## [Decision Letter · Decision Letter 0]

23 May 2023

PONE-D-23-08206Prediction of decreased mandibular bone density in familial Mediterranean fever via fractal dimension analysis: A cross-sectional retrospective studyPLOS ONE

Dear Dr. Nilüfer Ersan,

Thank you for submitting your manuscript to PLOS ONE. After careful consideration, we feel that it has merit but does not fully meet PLOS ONE’s publication criteria as it currently stands. Therefore, we invite you to submit a revised version of the manuscript that addresses the points raised during the review process.

We look forward to receiving your revised manuscript.

Kind regards,

Ewa Tomaszewska, DVM Ph.D

Academic Editor

PLOS ONE

Journal Requirements:

3. We note that Figure (2) includes an image of a [patient / participant / in the study]. 

If you are unable to obtain consent from the subject of the photograph, you will need to remove the figure and any other textual identifying information or case descriptions for this individual

Reviewers' comments:

Reviewer's Responses to Questions

**Comments to the Author**

1. Is the manuscript technically sound, and do the data support the conclusions?

Reviewer #1: No

2. Has the statistical analysis been performed appropriately and rigorously? 

Reviewer #1: Yes

3. Have the authors made all data underlying the findings in their manuscript fully available?

Reviewer #1: No

4. Is the manuscript presented in an intelligible fashion and written in standard English?

Reviewer #1: Yes

5. Review Comments to the Author

Reviewer #1: I have read carefully the manuscript entitled “Prediction of decreased mandibular bone density in familial Mediterranean fever via fractal dimension analysis: A cross-sectional retrospective study.” However, Authors didn't draft the paper well and I cannot recommend the publication of the article in its present form. The Authors should correct the manuscript before the re-submission of the work to Plos One.

First of all, the title does not correspond to the content of the manuscript, as no "prediction" or "bone density" has been established. The study only focuses on the assessment of the FD (fractal dimension) in different regions of the mandibular bone.

Secondly, there is no paragraph in which it is clearly stated that the method used to determine FD can be a measure of bone BMD (L66 and others). As indicated, FD is commonly used to assess the homogeneity of trabecular bone microarchitecture, not only in the mandible, and it is more accurately a measure of the regularity of the surface of the trabeculae. Therefore, measuring FD in the cortical bone area no longer aligns with the correct definition of FD provided in L67 and L206-208. Thus, additionally give additional references about measurements of FD of cortical bone.

Minor comments:

L56 In which country (countries) ?

L102 n- 25 in each group

L132 “grey scale”

L173 According to the data in the tables, all variables lack normal distribution. Am I right? Also information about checking the groups with the Chi-square test is missing.

Tables - correct the titles of tables 3-5, because you did not measure ROI, only FD in specific areas of the mandible

6. PLOS authors have the option to publish the peer review history of their article (what does this mean?). If published, this will include your full peer review and any attached files.

Reviewer #1: No

---

## [Author Response · Author response to Decision Letter 0]

3 Jun 2023

Jun 3, 2023

Manuscript Reference No: PONE-D-23-08206

Title: Prediction of decreased mandibular bone density in familial Mediterranean fever via fractal dimension analysis: A cross-sectional retrospective study

Dear Academic Editor and Reviewer(s),

Please find the responses to your comments and suggestions below. We are deeply appreciated that they assisted us to turn our manuscript to a far better version.

I have also made some additional corrections, such as grammatical or spelling errors and some other mistakes, that I have noticed during the revision of the manuscript. I would like to kindly ask you to consider them as a part of the revision process, as well.

Thank you for your time and considerations.

Kindest regards,

Dr. Nilüfer Ersan

 

Comments from the Academic Editor and Reviewer(s):

We do not have a laboratory protocol. However, we have deposited our study protocol along with the other supporting materials in https://doi.org/10.6084/m9.figshare.23259770. 

Revised accordingly.

We have shared our supporting materials in https://doi.org/10.6084/m9.figshare.23259770.

3. We note that Figure (2) includes an image of a [patient / participant / in the study]. 

We have successfully obtained consent from the individual and regarding this we inserted a statement in the Editorial Manager System and manuscript. We are keeping the file and did not submit it with the manuscript as instructed. 

 

Reviewers' comments:

Reviewer's Responses to Questions

Comments to the Author

1. Is the manuscript technically sound, and do the data support the conclusions?

Reviewer #1: No

There are no comments or suggestions to reply.

2. Has the statistical analysis been performed appropriately and rigorously?

Reviewer #1: Yes

There are no comments or suggestions to reply.

3. Have the authors made all data underlying the findings in their manuscript fully available?

Reviewer #1: No

We have shared our supporting materials in https://doi.org/10.6084/m9.figshare.23259770.

4. Is the manuscript presented in an intelligible fashion and written in standard English?

Reviewer #1: Yes

There are no comments or suggestions to reply.

5. Review Comments to the Author

Reviewer #1: I have read carefully the manuscript entitled “Prediction of decreased mandibular bone density in familial Mediterranean fever via fractal dimension analysis: A cross-sectional retrospective study.” However, Authors didn't draft the paper well and I cannot recommend the publication of the article in its present form. The Authors should correct the manuscript before the re-submission of the work to Plos One.

First of all, the title does not correspond to the content of the manuscript, as no "prediction" or "bone density" has been established. The study only focuses on the assessment of the FD (fractal dimension) in different regions of the mandibular bone.

The title was revised accordingly as ‘Fractal dimension analysis of different mandibular regions in familial Mediterranean fever patients: A cross-sectional retrospective study’.

Secondly, there is no paragraph in which it is clearly stated that the method used to determine FD can be a measure of bone BMD (L66 and others). As indicated, FD is commonly used to assess the homogeneity of trabecular bone microarchitecture, not only in the mandible, and it is more accurately a measure of the regularity of the surface of the trabeculae. Therefore, measuring FD in the cortical bone area no longer aligns with the correct definition of FD provided in L67 and L206-208. Thus, additionally give additional references about measurements of FD of cortical bone.

The manuscript was redrafted emphasizing the use of FD in the BMD evaluation and in assessment of cortical bone.

Minor comments:

L56 In which country (countries) ?

Related information was inserted in the text.

L102 n- 25 in each group

The phrase ‘in each group’ was inserted.

L132 “grey scale”

The term ‘grayscale’ was inserted to the statement.

L173 According to the data in the tables, all variables lack normal distribution. Am I right? 

In order to confirm the normality of the data, we have consulted to our statistician, who reevaluated the data in terms of different regions of interests and found that R3, R4, R5, L1, and L5 were normally distributed, while the others were not. As well, the data was reviewed and some minor corrections which did not affect the significance of the findings, were made. Abstract, Results section and related tables were revised accordingly. Thank you so much for your contribution.

Also information about checking the groups with the Chi-square test is missing.

Information regarding the Chi-square test was inserted to the Materials and methods section as: 'Chi-square test was used to assess the distribution of categorical data between FMF and control groups.’

Tables - correct the titles of tables 3-5, because you did not measure ROI, only FD in specific areas of the mandible

Titles of the Tables 3-6 were corrected.

6. PLOS authors have the option to publish the peer review history of their article (what does this mean?). If published, this will include your full peer review and any attached files.

Yes.

Do you want your identity to be public for this peer review? For information about this choice, including consent withdrawal, please see our Privacy Policy.

Reviewer #1: No

 There were no additional attachment files to consider.

I have arranged the figures in the PACE system. Nevertheless, I have sent an email to figures@plos.org and waiting for approval for the appropriateness of the figures. 

---

## [Decision Letter · Decision Letter 1]

22 Jun 2023

Fractal dimension analysis of different mandibular regions in familial Mediterranean fever patients: A cross-sectional retrospective study

PONE-D-23-08206R1

Dear Dr. Nilüfer Ersan,

We’re pleased to inform you that your manuscript has been judged scientifically suitable for publication and will be formally accepted for publication once it meets all outstanding technical requirements.

Kind regards,

Ewa Tomaszewska, DVM Ph.D

Academic Editor

PLOS ONE

Additional Editor Comments (optional):

Reviewers' comments:

Reviewer's Responses to Questions

**Comments to the Author**

1. If the authors have adequately addressed your comments raised in a previous round of review and you feel that this manuscript is now acceptable for publication, you may indicate that here to bypass the “Comments to the Author” section, enter your conflict of interest statement in the “Confidential to Editor” section, and submit your "Accept" recommendation.

Reviewer #1: All comments have been addressed

2. Is the manuscript technically sound, and do the data support the conclusions?

Reviewer #1: Yes

3. Has the statistical analysis been performed appropriately and rigorously? 

Reviewer #1: Yes

4. Have the authors made all data underlying the findings in their manuscript fully available?

Reviewer #1: No

5. Is the manuscript presented in an intelligible fashion and written in standard English?

Reviewer #1: Yes

6. Review Comments to the Author

Reviewer #1: I would like to thank the authors for reviewing and accepting all the comments and suggestions. In my opinion the article is now acceptable for publication.

7. PLOS authors have the option to publish the peer review history of their article (what does this mean?). If published, this will include your full peer review and any attached files.

Reviewer #1: No

---

## [Editor Report · Acceptance letter]

23 Jun 2023

PONE-D-23-08206R1 

Fractal dimension analysis of different mandibular regions in familial Mediterranean fever patients: A cross-sectional retrospective study 

Dear Dr. Ersan:

I'm pleased to inform you that your manuscript has been deemed suitable for publication in PLOS ONE. Congratulations! Your manuscript is now with our production department. 

Kind regards, 

on behalf of

Professor Ewa Tomaszewska 

Academic Editor

PLOS ONE